# Effects of Transcranial Direct Current Stimulation on Sleep in Athletes: A Protocol of a Randomized Controlled Trial

**DOI:** 10.3390/jcm11195883

**Published:** 2022-10-05

**Authors:** Mohammad Etoom, Mohammad Alwardat, Alia Alghwiri, Francesco Lena, Andrea Romigi

**Affiliations:** 1Physical Therapy Division, Allied Medical Sciences Department, Aqaba University of Technology, Aqaba 77110, Jordan; 2Department of Rehabilitation Sciences, Faculty of Applied Medical Sciences, Jordan University of Sciences and Technology, Irbid P.O. Box 3030, Jordan; 3Department of Physiotherapy, School of Rehabilitation Sciences, University of Jordan, Amman 11942, Jordan; 4Department of Physical Therapy, University of Pittsburgh, Pittsburgh, PA 15261, USA; 5IRCCS INM Neuromed, 86077 Pozzilli, IS, Italy; 6Department of Medicine and Health, University of Molise, 86100 Campobasso, Italy

**Keywords:** athletes, sleep wake disorders, transcranial direct current stimulation, randomized controlled trial

## Abstract

Background: Sleep disturbances are common among athletes. There is recently a growing interest in improving sleep quality by using noninvasive brain stimulation techniques such as transcranial direct current stimulation (tDCS). We hypothesized that bilateral dorsolateral prefrontal cortex anodal tDCS could improve sleep in different sports athletes. A randomized controlled trial is to be conducted to test this hypothesis. Methods: Eighty-four athletes are selected based on specific eligibility criteria and randomly allocated to the intervention or control group. Each participant will receive a 20-min session of bilateral anodal tDCS with an intensity of 1.5 mA (0.057 mA/cm^2^) in density 3 times a week for 2 weeks. The tDCS current will be delivered only for 30 seconds in the control group. This study’s outcome is a set of subjective and objective sleep parameters. Conclusion: This study assessed the effect of a novel tDCS protocol represented by bilateral anodal stimulation and may result in important advances in sleep management among athletes. Because of the high incidence and impact of athletes’ poor sleep quality, it is particularly important to explore effective interventions. Trial Registration: ClinicalTrials.gov: NCT05318352.

## 1. Introduction

Sleep disturbances are common among athletes [1]. The prevalence of high sleep disturbances is between 50 and 78% among elite athletes [2], which is more prevalent compared with non-athletic populations [3]. The sleep disturbances among athletes are generally characterized by poor sleep quality, daytime sleepiness, and habitual short sleep [2,4,5]. Different studies showed that disturbed sleep negatively affects overall athlete performance, anxiety, and physical and mental fatigue [6]. Sleep disorders are multifactorial, including sport and non-sport factors. The sport factors may include high training loads, unfamiliar sleeping environments, early morning training, long-haul travel, and night competitions [7,8,9,10]. On the other hand, individual chromatics, social demands, and habits may be considered non-sport factors [11].

A set of neuroimaging studies have identified a broad range of structural and functional changes in different brain areas in people with sleep disorders [12,13,14]. For example, people with insomnia and other sleep disorders exhibited decreasing metabolism in cortical-subcortical structures such as the bilateral prefrontal cortices, reticular activating system, and the left superior temporal, parietal, and occipital cortices [12,13,14]. Furthermore, the reduction in prefrontal cortex activity is directly linked to attentional reduction and cognitive flexibility impairments resulting from sleep insufficiency [15]. In this context, there is growing interest in the management of sleep disorders by utilizing invasive and noninvasive brain stimulation. Transcranial direct current stimulation (tDCS) is noninvasive brain stimulation that has been used in clinical trials to improve sleep in healthy and clinical populations [16].

tDCS may improve sleep indices through modulation of cortical activity and slowing oscillatory activity (<1 Hz) [17]. Slowing oscillatory activity is essential for the generation of oscillatory cycles that systematize brain activity during sleep [18]. It is considered the neurophysiological marker of slow wave activity (0.5–4.5 Hz) during the non-rapid eye movement (NREM) sleep stage [19]. The anodal stimulation for frontal areas can slow oscillation in the stimulated area and increase the activity of cortical and subcortical structures [17]. Therefore, we hypothesized that anodal stimulation of bilateral prefrontal cortices could improve sleep in athletes with poor sleep quality.

The current evidence shows some interventions inducing sleep in athletes [20]. Sleep education, sleep hygiene, nutritional plans, and encouraging short sleep nap opportunities are commonly used interventions in athletic populations [21,22]. However, adherence to those interventions in athletes is challenging. Furthermore, examining sleep in sports research has several limitations, such as inadequate experimental control, lack of females, and relying on self-reported sleep outcomes [1]. Therefore, there is a need for high-quality randomized controlled trials (RCTs) that assess the efficacy of alternative robust methods such as tDCS in different sports athletes. To our knowledge, there is only one clinical trial that examined the effect of tDCS on sleep in athletes and found improvement in some sleep parameters [23]. It included a small sample of athletes. We think that the effect of tDCS on sleep disorders in athletes is still unclear. Further robust studies are required to conclude the effect of tDCS on sleep for athletes. Hence, this RCT will aim to assess the effect of bilateral anodal tDCS over the right and left dorsolateral prefrontal cortices (DLPFCs) on a set of subjective and objective sleep parameters among athletes with poor sleep quality using a statistically powerful sample size.

## 2. Materials and Methods

### 2.1. Participants

Participant recruitment began in May 2022 and will continue until December 2023. Athletes from the national clubs and teams of different sports in Jordan will be invited to participate in this study. Elite athletes are eligible if they satisfy the following criteria: (1) being 16 years of age and older and (2) have a complaint of sleep impairment determined by a global score >5 on the Pittsburgh Sleep Quality Index (PSQI).

Exclusion criteria for the study will include any of the following: the use of sleep medications or other interventions interfering with sleep and sleepiness, more than one concussion in the past year, using a pacemaker, pregnancy, diagnosis of migraines with a high frequency of episodes, wearing a metal implant, or epilepsy. 

### 2.2. Study Design

The study design is a prospective, double-blinded RCT. It will be reported according to the Consolidated Standards of Reporting Trials (CONSORT) statement for non-pharmacological treatment [24]. The study was approved by the institutional review board council at the University of Jordan (80-2021, 11 August 2021). The clinical trial is registered with the clinicaltrials.gov web site (registration number: NCT05318352). All participants will sign written informed consent prior to participation according to the Declaration of Helsinki.

### 2.3. Randomization Allocation and Blinding

Participants will be allocated consecutively to randomization using the tool from the randomization.com website (http://www.randomization.com, Accessed date: 10 May 2022) by a researcher who is not in contact with any participant. The investigators will use the suggested first generation for RCT studies. The included participants will be randomly allocated to the parallel tDCS or control groups in a balanced allocation ratio of 1:1. Allocation concealment will continue until the end of the study.

All assessment, evaluation, and intervention procedures will be carried out by examiners who will be blinded to the allocation of the participants. A trained research assistant that is not otherwise connected with the study will set the stimulation characteristics as indicated by the information in a sealed envelope opened upon the participants’ visits (real or sham). The mode of stimulation will not be revealed to the participants or investigators.

### 2.4. Sample Size Calculation

The sample size was calculated using G*Power software according to the study of Charest et al. (2021) [23]. The study found d = 0.7012 for the Cohen’s effect size of tDCS on PSQI as the screening outcome of this study. Based on a 0.7012 effect size, bidirectional alpha of 0.05, and 80% test power, a minimum of 70 participants are required for this study. The sample will be increased by 20% to compensate for possible dropouts, and an overall sample of 84 participants will be included in this study.

### 2.5. Procedures

The research procedure will be explained to eligible athletes if they agree to participate. Informed consent will be obtained from each participant. Figure 1 demonstrates the CONSORT flow diagram for this study. Baseline assessments (T0) will be conducted for demographic, health, and sport-related information, clinical characteristics, and primary and secondary outcome measures at baseline. Post intervention assessments (T1) for comparing the immediate efficacy between the two groups of primary and secondary outcome measures will be performed directly after completion of the intervention protocol. Follow-up assessments (T2) will be conducted 1 month after the end of the intervention.

### 2.6. Assessments and Outcome Measures

Demographics include the date of birth, educational level, marital status, weight, and height. Health-related questions concern the diagnosis of any other disorders and current medications used. Sport-related information that will be gathered includes the type and level of sport, number of years in practice, number of times they practice per week, and for how long.

All assessment instruments will be used in their validated Arabic versions. The primary and secondary outcome measures include both subjective and objective sleep outcomes.

### 2.7. Primary Outcome Measures

#### 2.7.1. A-Sleep Monitoring

ActiGraph via synchronized wGT3X-BT software [25] will be used to objectively assess the physical activity parameters and sleep parameters. Actigraph is a valid and reliable tool for assessing sleep in healthy individuals [26]. The Actigraph sleep parameters are the following:i.Time in bed: the time between the ‘Lights Out’ and ‘Got Up’ times;ii.Actual sleep time: the total time spent sleeping according to the epoch-by-epoch wake or sleep categorization, excluding sleep latency and waking after sleep onset;iii.Wake after sleep onset: the amount of time a person spends awake, starting from when he or she first falls asleep to becoming fully awake and not attempting to go back to sleep;iv.Sleep efficiency: the ratio between the actual sleep time and time in bed.

#### 2.7.2. B-Insomnia Severity Index (ISI)

The ISI will be used to quantify the severity of sleep impairments during the nights of the previous month [27]. ISI is a self-reported questionnaire that includes seven questions. The total score of the ISI ranges between 0 and 28, with a higher score indicating more severe insomnia.

#### 2.7.3. C-Epworth Sleepiness Scale (ESS)

The ESS is used to assess the level of daytime sleepiness [28]. The ESS is a self-administered eight-item questionnaire that asks about the chance of falling asleep during performing eight daily activities on a scale from 0 to 3. The total score ranges from 0 to 24, with a higher score indicating worse daytime sleepiness.

### 2.8. Secondary Outcome Measures

#### 2.8.1. A-PSQI

PSQI is a self-reported questionnaire with 19 questions that measure various aspects of sleep quality [29]. The response to the questionnaire is rated on a scale from 0 to 3, with 0 indicating no sleep difficulty and 3 indicating severe sleep difficulties. The PSQI global score ranges from 0 to 21, with a higher score indicating poor sleep quality. Individuals with a PSQI global score of >5 can be classified as having poor sleep quality. The PSQI has good psychometric properties in sport populations [30].

#### 2.8.2. B-Depression Anxiety Stress Scale-21 (DASS-21)

The DASS is composed of 21 items that assess the psychological symptoms of depression, anxiety, and stress [31]. The response to DASS items involves using a Likert scale that ranges between 0 and 3, with a higher score indicating a worse psychological status. It has been recommended to use the DASS-21 with athletes over other measures of mental health [32].

#### 2.8.3. C-Medical Outcome Study Short-Form Survey 12-Item (SF-12)

The SF-12 is a self-reported questionnaire that assesses the impact of health on an individual’s quality of life [33]. The SF-12 consists of 8 domains: physical function, role (physical), bodily pain, general health, vitality, social functioning, role (emotional), and mental health. The possible total scores range between 0 and 100, with higher scores demonstrating better quality of life [33].

### 2.9. Intervention

Depending on the randomization allocation, participants will receive either a real tDCS or sham tDCS protocol for three sessions per week for 2 weeks, with a total of six sessions. 

### 2.10. Treatment Group (Real tDCS Group)

Direct current will be delivered from a battery-driven constant current stimulator (Neuro-electric, NIC, Barcelona, Spain). Each participant will receive 20 min of bilateral anodal tDCS (positively charged electrode). Anodal electrodes are placed over the right and left DLPFC (F3 and F4) according to the 10–10 international EEG system [34] (Figure 2) with small-size (1 cm^2^) sponge electrodes soaked in saline with an intensity of 1.5 mA (0.057 mA/cm^2^ in density). This will be repeated six times during the 2 weeks. The bilateral cathodes (negatively charged electrodes) will be placed over the left and right supraorbital areas as reference codes (FP1 and FP2).

All participants will be asked to remain minimally active during this time and engage in either light watching of television or listening to the radio.

### 2.11. Control Group (Sham tDCS Group)

For sham stimulation, the electrodes will be placed in the same positions as for the real tDCS stimulation. However, a constant current of 1.5 mA will only be delivered for 30 s (30-s ramp on). This stimulation protocol will provide an initial period of a tingling sensation similar to that one would perceive during the real tDCS protocol. The data and instructions on the device display will not differ from the real tDCS settings.

### 2.12. Statistical Analysis

The participants’ characteristics and outcome measures at baseline will be compared between the two groups using the Kruskal–Wallis test. Data will be screened for missing data, normality, and outliers. For variables that are not normally distributed, non-parametric tests will be used. The normally distributed data will be analyzed by repeated measures analysis of variance (ANOVA) to evaluate the therapeutic effect and changes between the two groups and within groups (group: sham vs. real tDCS) × (time: pre- vs. post-intervention vs. 1-month follow-up). Post hoc analysis will be applied to determine the exact groups with significant changes if significant findings appear. For the variables not normally distributed, we will use Friedmann’s test. The correlation between the changes in the Actigraph results with the athletes’ characteristics, DAAS-21, SF-12 pain, general health, role (emotional), and mental health domains at baseline will be analyzed by repeated measures linear regressions to understand the mechanism of improvement. The data will be analyzed using SPSS software, and a conservative *p*-value of 0.05 will be considered.

## 3. Discussion

The planning of sleep interventions for athlete populations is different than for other populations, such as as neurological or psychological populations. It requires special considerations for the following reasons. First, the prevalence of sleep disturbances in athletes is high. Second, sleep medication can negatively influence sports performance by impairing the efficacy of resistance, prolonged endurance, and high-intensity exercises [35]. Finally, the current interventions for sleep require a long time and commitment, making adherence challenging [1]. Therefore, we suggest that using a robust intervention method such as tDCS is more feasible for athletes.

This study will assess the effect of bilateral anodal tDCS stimulation specifically over the right and left DLPFCs through small electrodes. The only tDCS study [23] that was conducted for athletes reported that the stimulation was for frontal areas without specifying the targeted areas. The tDCS programs for sleep were generally for the left DLPFC in other populations [19]. The selected tDCS stimulation protocol (bilateral anodal) to improve sleep quality is novel in athletes.

The bilateral anodal tDCS could improve sleep via direct and indirect effects on sleep structures [16]. The right and left prefrontal cortices have important roles in sleep regulation and showed decreased activity in people with poor sleep quality and insomnia [36]. Bilateral DLPFC anodal stimulation might induce the ‘top-down’ pathway of sleep–wake regulation [37]. DLPFC networks generate slow oscillations that are biphasic changes in the membrane potential from a hyperpolarized state (down phase) to a depolarized phase (top phase) (<1 Hz) [38]. Furthermore, tDCS could increase the NREM2 sleep stage and decrease slow wave sleep stage regulation [18]. In addition to the direct effects, the tDCS could improve pain, depression, or anxiety and therefore sleep [16]. To understand the mechanism of improvement, correlation analysis of the changes in objective sleep results with athletes’ characteristics and secondary outcomes at baseline will be conducted.

In this protocol, tDCS parameters of a 1.5-mA intensity in 1 cm^2^ (0.057 mA/cm^2^ in density) for 20 min/session 6 times for 2 weeks will be used. The rationale for using 1.5 mA is to generate slow oscillation in the DLPFCs. The tDCS intensity ranges between 1 and 2 mA in sleep trials [16]. A previous study found that 1.5 mA is more effective than 1 mA in people with poor sleep quality [39]. The current parameters are considered safe based on data of more than 33,200 sessions and 1000 participants [40]. The study concluded that the tDCS protocols at ≤4 mA for ≤40 min had not produced any reports of a serious adverse effect or permanent injury [40].

The current study’s outcome measures include subjective and objective sleep measurements, psychological outcomes, and quality of life, which were not previously examined in athletic participants. A previous study [26] found low correlation between the subjective and objective sleep measures and recommended using subjective and actigraphy sleep measures to comprehensively assess sleep. Therefore, exploring the therapeutic effects of the bilateral anodal tDCS on different sleep parameters, quality of life, and psychological status in the athletic participants with sleep disorders may provide a promising noninvasive tool for athletic sleep disorder management.

## 4. Conclusions

The results of this trial may provide an important advancement and contribution in sleep management among athletes. Because of the high incidence and impact of poor sleep quality among athletes, it is particularly important to explore an effective method for sleep disorder management in athletes. This experiment will provide a new direction in sleep management among athletes.

## Figures and Tables

**Figure 1 jcm-11-05883-f001:**
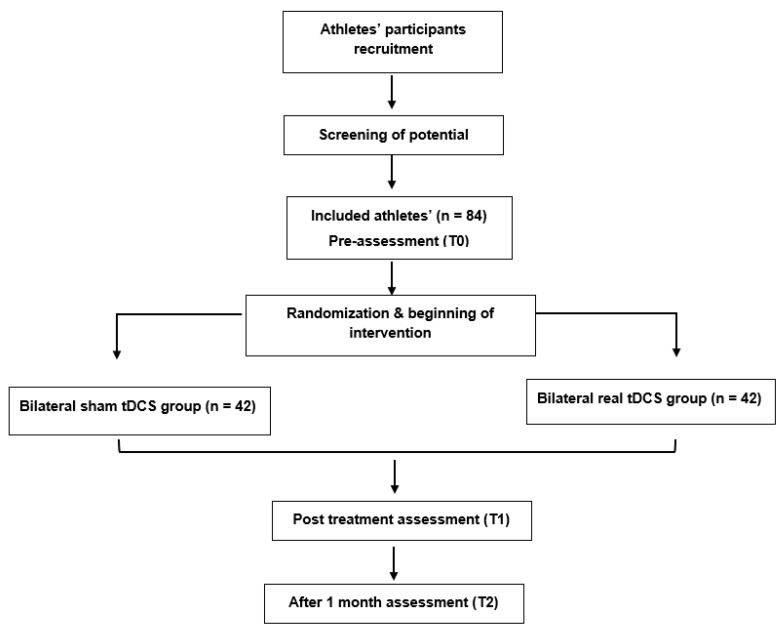
The CONSORT flow diagram of the study procedures. CONSORT = Consolidated Standards of Reporting Trials, tDCS = transcranial direct current stimulation.

**Figure 2 jcm-11-05883-f002:**
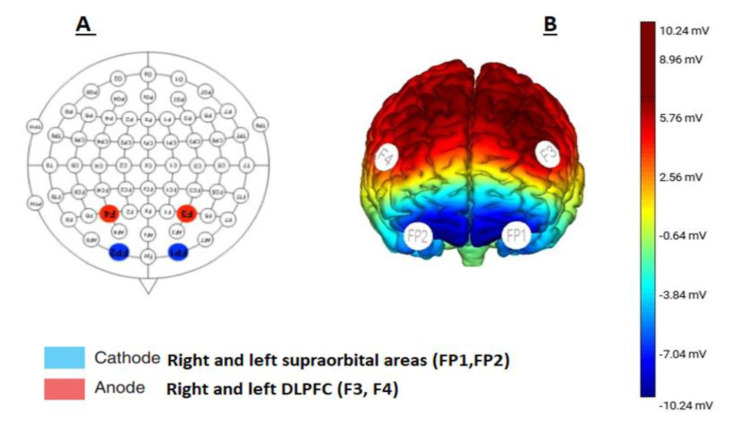
The tDCS electrode placement. (**A**) Anodal and cathodal stimulation according to the 10−10 system. (**B**) Brain schematic view showing the targeted cortices through the anodal and cathodal brain stimulation.

## Data Availability

The data presented in this study will be available on request from the corresponding author.

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
