# Peer review of "Effects of Transcranial Direct Current Stimulation on Sleep in Athletes: A Protocol of a Randomized Controlled Trial"

_jcm, 2022, doi:10.3390/jcm11195883_

Round 1

Reviewer 1 Report

Dear Authors,

your protocol is interesting;  it studies the possibility of improving the sleep of athletes by means of transcranial direct current stimulation (tDCS). In my opinion, considerations should be added for possible damage to brain tissue by transcranial stimulation to assure more athletes’ safety.

Reviewer 2 Report

Dear Authors,

The manuscript entitled “Effects of transcranial direct current stimulation on sleep in athletes: A protocol of a randomized controlled trial” aims to assess the effect of bilateral anodal tDCS over the right and left dorsolateral prefrontal cortex (DLPFC) on a set of subjective and objective sleep parameters among athletes with poor sleep quality using a statistically powerful sample size.

Strengths of the study include the authors’ care by submitting the study to CONSORT and adopting an electronic randomization tool. However, some details about typo errors and methodological questions are detailed here.

1.      Minor edits

1.1.            Line 51, page 2: Transcranial direct current stimulation (tDCS) is a 50 non-invasive brain stimulation that has been used in clinical trials to improve sleep in healthy 51 and clinical populations.

1.2.            Line 123, page 3: compare the immediate efficacy0 between the two groups on primary and secondary

1.3.         Lines 185-185, page 5:

1.4.            Line 216, page 6: the prevalence of sleep disturbances in athletes is high.

1.5.            Line 220, page 6: we suggest the use of … or we suggest that using a robust…

 2.      General comment

2.1.            The introduction session is clear and provides the reader with all the information necessary to understand the rationale of the study.

2.2.            Verbal tense varied a lot between past and future, and I got a little bit confused about the data collection’s realization.

2.3.            Besides verbal tense, the fact that the authors added an “n” (equal to 48), different from the n suggested by sample size calculation, in the CONSORT flow chart also contributed to confusing the reader about whether data collection had been already realized.

3.      Methods

3.1.            The method section is clear; however, an important methodological issue was not cited. Authors should that groups will show the same sleep quality level on the pretest moment to avoid bias when comparing groups.

4.      Statistical Analysis

4.1.            The non-parametric test equivalent to ANOVA is Friedmann’s Test. Why did you choose Mann-Whitney Test?

5.      Discussion

5.1.            The authors reported the study's aim and appropriately justified their choice to stimulate the selected areas. They also pointed out a limitation of the only tDCS study that was conducted with athletes, and briefly explained possible mechanisms behind the effects of tDCS on sleep. In my opinion, the discussion is short, and the authors could spend more time explaining those mechanisms to clarify them.

6.      Conclusion

6.1.            The conclusions are correct, not speculative, and summarized the main points reported in the study.

Reviewer 3 Report

The authors focused on tDCS as a noninvasive intervention for sleep in athletes. 

pg 5, line 178: Consider explaining the reason for using 1.5 mA versus 1 or 2 mA. Use supportive literature here.

pg 2, line 83-86: The exclusion criteria should be clearer. Please provide information about exclusion of individuals with sleep disorders, stroke, etc. Was a there a screening for sleep disorders such as the SCISD to r/o risk for sleep apnea?  Was concussion the only exclusion related to history of head trauma? If so, please discuss this limitation in the discussion section.

pg 4, line 141: Please include information on the reliability of the Actigraph.

pg 5, line 175: Please include more information on SF 12. How many items are on the survey, range of scores, and what do higher scores indicate?

pg 6, line 214 Discussion: The authors should discuss the current literature related to use of tDCS for sleep and related conditions. Since there are no results, the authors should focus on how the literature may or may not support their hypothesis. It is worth noting that athletes may experience other sleep-related conditions (e.g., pain/injuries/psychological disorders) that may interfere with the outcome. This should be acknowledged in the discussion section. In addition, discuss the literature as it relates to potential indirect effects of tDCS on sleep via other mechanisms such as depressive symptoms/chronic pain.

Round 2

Reviewer 1 Report

no comment